# A Piezoelectric MEMS Speaker with a Combined Function of a Silent Alarm

**DOI:** 10.3390/mi14030702

**Published:** 2023-03-22

**Authors:** Qi Wang, Tao Ruan, Qingda Xu, Zhiyong Hu, Bin Yang, Minmin You, Zude Lin, Jingquan Liu

**Affiliations:** 1National Key Laboratory of Science and Technology on Micro/Nano Fabrication, Shanghai Jiao Tong University, Shanghai 200240, China; 2Department of Micro/Nano-Electronics, Collaborative Innovation Center of IFSA, Shanghai Jiao Tong University, Shanghai 200240, China

**Keywords:** piezoelectric, micro-electro-mechanical system (MEMS), rigid–flexible-coupling, speaker, silent alarm

## Abstract

To explore the versatility of speakers, a piezoelectric micro-electro-mechanical system (MEMS) speaker combining the function of a silent alarm is proposed, which mainly comprises a lead zirconate titanate (PZT) actuation layer and a rigid–flexible coupling supporting layer. Measurements performed on encapsulated prototypes mounted to an artificial ear simulator have revealed that, compared to a speaker with a rigid supporting layer, the sound pressure level (SPL) of the proposed piezoelectric MEMS speaker with a rigid–flexible coupling supporting layer is significantly higher and is especially higher by 4.1–20.1 dB in the frequency range from 20 Hz to 4.2 kHz, indicating that the rigid–flexible coupling supporting layer can improve the SPL significantly in low frequency. Moreover, the spectral distribution characteristic of its playback audio is similar to that of the commercial electromagnetic type. The device can also function as a silent alarm based on oral airflows in dangerous situations, as it performs well at recognizing words according to their unique voltage-signal characteristics, and can avoid the effects of external sound noise, body movement, long distance, and occlusion. This strategy provides inspiration for functional diversification of piezoelectric MEMS speakers.

## 1. Introduction

Over recent years, there has been much research interest in audio electronics, such as micro speakers [1,2], hearing aids [3,4], and speech recognition sensors [5,6,7]. These single-function devices have greatly improved people’s quality of life, but they are still far from meeting the ever-increasing demands for convenience. Therefore, to avoid the inevitable heaviness and messiness caused by wearing various single-function devices, dual function or multifunction wearable devices have become desirable and constitute new research trends [8,9,10,11,12]. In particular, the rapid development of micro-electro-mechanical system (MEMS) technology is expected to realize the low-cost mass production of such devices.

Existing speakers are mainly divided into electromagnetic, electrostatic, and piezoelectric types according to the working mechanism. The traditional electromagnetic speaker based on electromagnetic induction still cannot be replaced by the MEMS one because of its relatively large permanent magnet [13,14]. Despite offering benefits such as low membrane mass and ease of integration, electrostatically-driven capacitive MEMS speakers are still subject to limitations such as the pull-in effect and the requirement for high driving voltage, as highlighted in [15,16]. Piezoelectric MEMS speakers based on the reverse piezoelectric effect are gradually attracting attention for the advantages of low power consumption, high theoretical sound pressure level (SPL), fast response, as well as being dust-free, and viable for mass production. Therefore, piezoelectric materials provide a potential solution for MEMS speakers [17,18,19]. Researchers have developed various piezoelectric materials, such as ZnO [20,21], AlN [22], PZT [23,24], PMN-PT [25,26,27], and PZN-PT [28]. The present choice of sputtering piezoelectric PZT thin film as the preferred material is related to not only our material preparation process but also the fact that PZT has the advantages of higher piezoelectric charge constant and electromechanical coupling coefficient [29,30,31].

Although researchers have adopted new working principles, structures, and driving electrodes [2,29,32,33], piezoelectric MEMS speakers still suffer from the challenge of low SPL, and therefore cannot realize multifunction. In our previous work, the obtained SPL is still small when the radius is 1.5 mm [32], which needs to be further considered to reduce structural stiffness. In addition, existing speech-recognition devices are either disturbed by external sound noise [34], body movement [35], or are limited by distance or occlusion [6,7], thereby leading to errors or failures in speech recognition. These problems have been encouraging researchers to find a more accurate and convenient way of the silent speech recognition system. This work would gain further significance if the aforementioned piezoelectric MEMS speaker, which currently serves a single function, could also function as a silent alarm.

Herein, a piezoelectric MEMS speaker combining function of silent alarm is demonstrated. The deposition of 10 μm thick parylene C on the upper surface before etching the back cavity is the key process when forming the rigid–flexible coupling supporting layer. Moreso, the rigid vibration membrane with a maximum radius of 2 mm is a central circle surrounded by concentric double rings. Compared to a speaker with a rigid supporting layer, this proposed piezoelectric MEMS speaker with rigid–flexible coupling supporting layer can improve the SPL significantly in the frequency range from 20 Hz to 4.2 kHz. Moreover, in the frequency range of 20 Hz to 20 kHz, a volume higher than 72.3 dB SPL is achieved at driving voltage of 10 V, which satisfies the basic needs of human hearing. The proposed device can also function as a silent alarm based on oral airflow, which performs well at identifying an alarm silently through its unique voltage-signal characteristics, and avoids the effects of external sound noise, body movement, long distance, and occlusion. This proposed device provides inspiration for expanding the capabilities of piezoelectric MEMS speakers.

## 2. Strategy and Design

The strategy and design of the piezoelectric MEMS speaker combining function of silent alarm are shown in Figure 1a,b. It comprises a vibration membrane with a central circle surrounded by concentric double rings and a flexible parylene C supporting layer. The shape and rigid–flexible coupling structure of the vibration membrane were designed to reduce structural stiffness, which can increase the vibration displacement, thereby facilitating the increase in SPL [32] or voltage signal for a given oral airflow blowing onto the surfaces [36]. Moreover, the parylene C was deposited on the device surface and back cavity to form the rigid–flexible coupling supporting layer and for avoidance of the influence of water vapor on its performance. The proposed device can function as a speaker by using the inverse piezoelectric effect of the PZT thin film and can also perform well at recognizing words according to their unique voltage-signal characteristics when spoken silently. As shown in Figure 1c, the size of the chip is less than 10 mm, and the shape of the vibration membrane can be clearly displayed. The functions of the proposed device are shown in Figure 1d.

## 3. Materials and Methods

The preparation process of the chip is shown in Figure 2a. The thickness of the buried oxygen layer SiO_2_ was ~1.5 μm, the thickness of the top Si layer on the SOI (silicon on insulator) wafer was ~2 μm, and ~500-nm-thick SiO_2_ was deposited on the top Si surface by plasma-enhanced chemical vapor deposition. Finally, a ~100-nm-thick Pt layer, a ~1-μm-thick PZT piezoelectric thin film was deposited by RF magnetron sputtering (piezoelectric coefficient (d_31_) = ~−220 pm/V, dielectric constant (ε_r_) = ~900, and Young’s modulus = ~70 × 10^9^ Pa), and a ~100-nm-thick Pt layer were sputtered successively on the deposited SiO_2_ surface (Figure 2(a1)). The sputtered PZT does not require a dedicated polarization due to the self-polarization effect.

Then, the upper electrode Pt layer was etched to form a central circle surrounded by concentric double rings (Figure 2(a2)) (see Appendix A for detailed sizes), then BHF was prepared in the volume ratio of V[NH_4_F(40%)]: V[HF] = 1:5, and the etching solution was obtained by mixing a solution with a volume ratio of V[BHF]: V[HCl]: V[H_2_O] = 1:25:175. The etching residue attached to the PZT surface was removed after etching the 1-μm-thick PZT ((Figure 2(a3)) [37,38]. The process of etching the lower electrode was the same as that for the upper electrode (Figure 2(a4)). The deposited SiO_2_, top Si, middle buried oxygen layer SiO_2_ were then etched in sequence (Figure 2(a5)). Then, ~10 μm parylene C was deposited on the membrane surface (Figure 2(a6)). Then, the back cavity was formed after etching barrier SiO_2_ (Figure 2(a7)). Finally, a ~0.5-μm-thick layer of parylene C was deposited on the device surface and back cavity to avoid the influence of humidity on device performance by isolating water vapor (Figure 2(a8)) [39,40,41]. Of course, the functions of speaker and silent alarm are exclusive, so that a circuit rotation device and a wireless data transmission module need to be built to facilitate the conversion of functions and data transmission. However, these efforts are still in the research process due to the current limited knowledge in this area.

The SEM images were characterized by a scanning electron microscope (FE-SEM, ZEISS Gemini, Oberkochen, Germany). Impedance Analyzer (KEYSIGHT, E4990A, Santa Rosa, CA, USA) was used to measure the relationship between the frequency and impedance or phase angle, respectively. The SPLs of the piezoelectric MEMS speakers were characterized by an audio analyzer (FX100, NTi Audio, Schaan (Liechtenstein), Switzerland). The SPLs distributions at each moment were recorded by an audio and acoustic analyzer (XL2, NTi Audio, Schaan (Liechtenstein), Switzerland). The acoustic tests were all carried out in an anechoic box. The output voltage signals and voltage-signal characteristics of the device were recorded by a digital storage oscilloscope (DSO-X 2024A, California, Agilent). Parylene C was deposited by using a parylene C thin-film deposition system (PDS 2010; SCS, Indianapolis, IN, USA).

## 4. Discussions

### 4.1. Photograph and SEM Images

Figure 2b shows the device encapsulated with a 3D-printed shell. Figure 2c shows a cross section of the functional material layer PZT with thickness of ~1 μm. As can be seen in Figure 2d, the vibration membrane is patterned into a central circle surrounded by concentric double rings. Figure 2e depicts the parylene C deposited on the vibration membrane surface. Figure 2f shows the inclined etching gap of the vibration membrane, with a gap width of ~12 μm. Moreso, the side of a ~10-μm-thick parylene C can be clearly demonstrated, as shown in Figure 2g.

### 4.2. Measured SPL

Compared with the same thick rigid (SiO_2_/Si) supporting layer, the rigid–flexible coupling one can greatly reduce structural stiffness, thereby increasing the vibration displacement and SPL.

Figure 3a shows that the SPL of the proposed piezoelectric MEMS speaker with a rigid–flexible coupling supporting layer is significantly higher than that with rigid supporting layer, and especially, is higher by 4.1–20.1 dB in the frequency range from 20 Hz to 4.2 kHz, which indicates the rigid–flexible-coupling supporting layer can improve the SPL significantly in low frequency. Moreover, a figure higher than 62 dB SPL is obtained in the frequency range of 20 Hz–20 kHz, which is at least 10 dB higher than the previous work [32], thus meeting the basic needs of human hearing [42]. Besides, an SPL of ~102.7 dB was achieved at a resonance frequency of 4 kHz. Compared with the speaker with a rigid supporting layer, the structural stiffness of the rigid–flexible coupling supporting layer prepared by us is greatly reduced, thereby making the vibration offset displacement of the vibration membrane higher, which can greatly improve the SPL at low frequency. As shown in Figure 3b, in the human audible frequency range of 20 Hz–20 kHz, a volume higher than 62 dB SPL is obtained at 2 V, and greater than 72.3 dB SPL is achieved at driving voltage of 10 V. With increasing driving voltage, the highest SPL at the resonance frequency of ~4 kHz changes from ~102.7 dB (2 V) to ~107.5 dB (4 V) to ~110.7 dB (6 V) to ~111.6 dB (8 V) and finally to ~112.5 dB (10 V). 

As shown in Figure 4a–d, the SPL distribution and effective SPL at each moment when the piezoelectric MEMS speaker and a commercial electromagnetic one are used to play the same song. As can be seen, the effective SPL (L_Aeq_) of the prepared piezoelectric MEMS speaker at each moment is 3–8 dB lower than that of the commercial one with the same size, and its maximum (L_AFmax_) and minimum (L_AFmin_) effective SPLs are also lower than those of the commercial one by 3–8 dB. However, compared with piezoelectric MEMS speaker, electromagnetic ones are difficult to miniaturize because of their larger permanent magnets [2,13,14], and also the performance of the prepared piezoelectric MEMS device meets the basic hearing needs for music, broadcast, voice interaction, etc.

### 4.3. Silent Alarm in Dangerous Situations

In actuality, it may not be feasible for individuals to vocalize for assistance or raise an audible alarm when confronted with a potential kidnapping or involuntary confinement, as doing so may provoke hostile parties. Thus, the development of a silent alarm sensor represents a meaningful and promising technological advancement [43,44]. In order to expand the application of piezoelectric MEMS speaker, this proposed device was attempted to be used as a silent alarm sensor, and the working mechanism is shown in Figure 5a. If we speak silently, different words produce different oral airflows that impact the membrane surface of the proposed piezoelectric MEMS device and produce different deformations, thereby generating different voltage signals. Therefore, different words can be recognized according to their unique voltage-signal characteristics. Of course, temperature also affects the voltage signals, and the increase or decrease in temperature are consistent to that of oral airflow speed, so their waveform variation law is considered to be the same. The voltage signal generated by this structure is significantly larger than the previous design work, which contributes to the accuracy of identification (Appendix A) [36]. The distance between the mouth and the device is within ~2 cm. The corresponding valley value and number as well as overall waveform are the voltage signal characteristics being preliminary considered. Less than 2 cm is chosen as the test distance because the voltage signal generated is relatively obvious, as shown in Figure 5b. As shown in Figure 5c, for words “silent”, the voltage-signal characteristics produced by the device when speaking silently with body movement are consistent with those without movement, which shows that the performance of the device is unaffected by body-movement interference. When different people speak the same word silently, the voltage-signal characteristics are consistent, which indicates the device’s universal applicability to different people, as shown in Figure 5d. As can be seen in Figure 5e, when the same word is said silently, the voltage signal characteristics are also consistent in quiet and noise environments, which shows the good anti-interference performance of this device in a noisy sound environment. In Figure 5f, the words “alarm”, “danger”, “save” and “me” all conform to the above rules, which also well proves the universal applicability of this method. When speaking continuously, the dangerous signals are well recognized through their unique voltage signal characteristics obtained, which proves the feasibility of the device in practical application (Appendix A). The above are only our preliminary mechanism analysis and exploration of the piezoelectric MEMS silent alarm sensor based on oral airflow, but it can also preliminarily reflect the feasibility of our proposed scheme and puts forward a new possible alternative for the traditional alarm system. Besides, this silent speech-recognition sensor can avoid the effects of external sound noise, body movement, long distance, and occlusion compared to the state of the art. Future work will combine machine learning to show remotely what they say [45,46], which is expected to be used to save people in dangerous situations, even to serve people with acquired throat injury or serious diseases, weak bodies, or who are unable to speak loudly or are in a public quiet environment. Thus, this piezoelectric MEMS speaker can also be used as a silent alarm sensor. However, we just elaborated on the working mechanism and made an attempt on its possibility, and the subsequent detailed studies are still ongoing.

### 4.4. Compared with Previous Works

The comparisons between this work and our previous works are shown in Table 1, showing this work has its differences and advantages compared to previous works.

## 5. Conclusions

In conclusion, a piezoelectric MEMS speaker combining function of silent alarm was reviewed. The speaker mainly comprises a PZT actuation layer and a rigid–flexible coupling supporting layer. The deposition of 10 μm thick parylene C on the upper surface before etching the back cavity is the key process in forming the rigid–flexible coupling supporting layer. Additionally, the rigid vibration membrane is a central circle surrounded by concentric double rings. This proposed piezoelectric MEMS speaker, which features a rigid–flexible coupling supporting layer, can enhance the 4.1–20.1 dB SPL at low frequencies ranging from 20 Hz to 4.2 kHz, as compared to a speaker with a rigid supporting layer. In addition, this can also improve the distribution of SPL and maintain an effective SPL similar to that of a commercial electromagnetic speaker when playing the same song. The device is also capable of serving as a silent alarm in hazardous situations by detecting oral airflows. It performs exceptionally well in silently recognizing alarms based on their distinctive voltage-signal features, while effectively avoiding the impacts of external noise, body movement, long distance, and blockage. Thus, both the piezoelectric and inverse piezoelectric effects of PZT thin film can be utilized. This prepared device provides a reference for the functional expansion of piezoelectric MEMS speakers.

## Figures and Tables

**Figure 1 micromachines-14-00702-f001:**
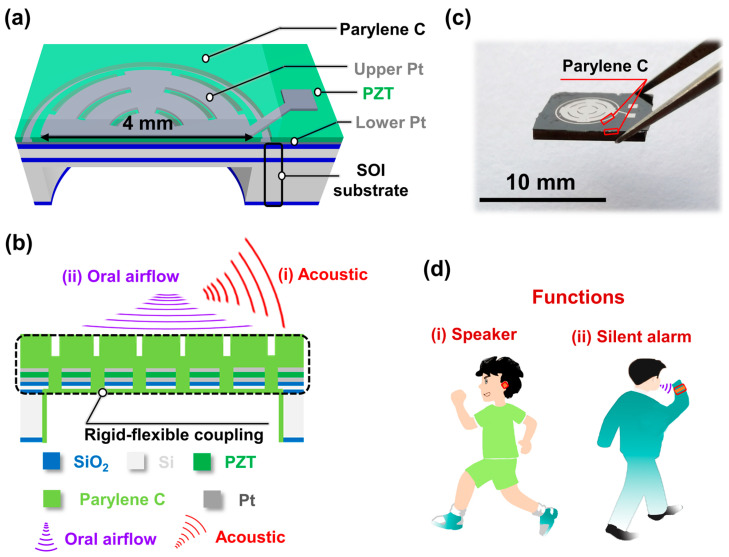
Strategy and design of piezoelectric MEMS speaker combining function of silent alarm. (**a**,**b**) Strategy and design, (i) Acoustic, (ii) Oral airflow. (**c**) Photographs of chip clamped by tweezers. (**d**) Piezoelectric MEMS speaker (i) combining function of silent alarm (ii).

**Figure 2 micromachines-14-00702-f002:**
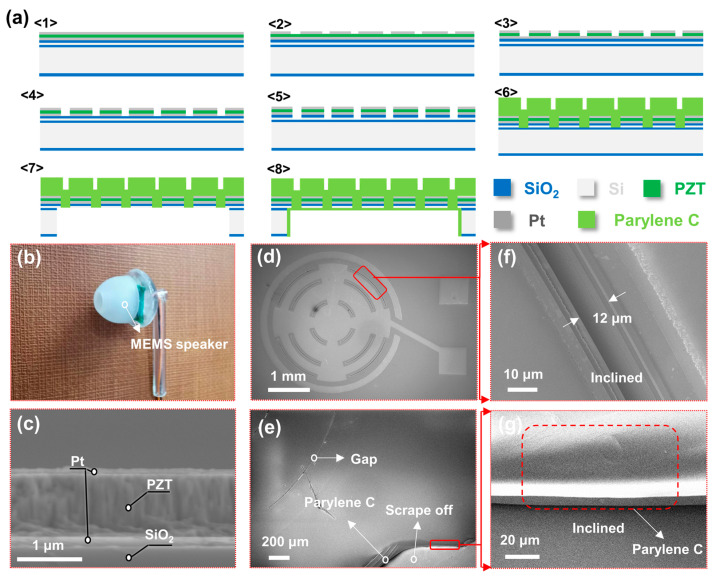
Preparation process and SEM images of the piezoelectric MEMS speaker combining function of silent alarm. (**a**) Process of fabricating the proposed device: (1) materials, (2) etching the upper electrode, (3) wet etching the PZT, (4) etching the lower electrode, (5) etching gaps on the upper surface, (6) depositing thick parylene C, (7) etching the back cavity, (8) depositing thin parylene C. (**b**) Encapsulated speaker. Scanning electron microscopy (SEM) images of (**c**) PZT, (**d**) the whole surface of the device without parylene C, and (**e**) the deposited parylene C, and (**f**) the inclined gap, and (**g**) the inclined view of parylene C section.

**Figure 3 micromachines-14-00702-f003:**
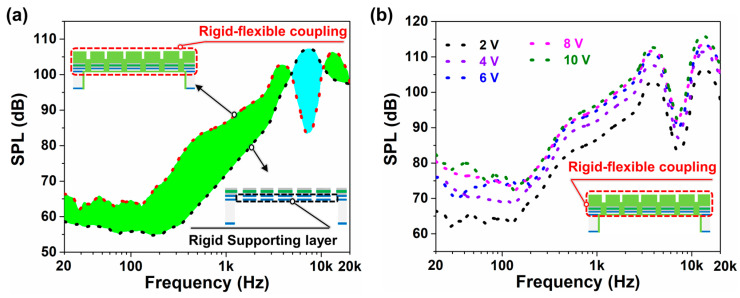
Measured SPL. (**a**) Comparison of SPL of a piezoelectric MEMS speaker with rigid supporting layer and rigid–flexible coupling one. (**b**) Measured SPL of piezoelectric MEMS speaker at different driving voltages.

**Figure 4 micromachines-14-00702-f004:**
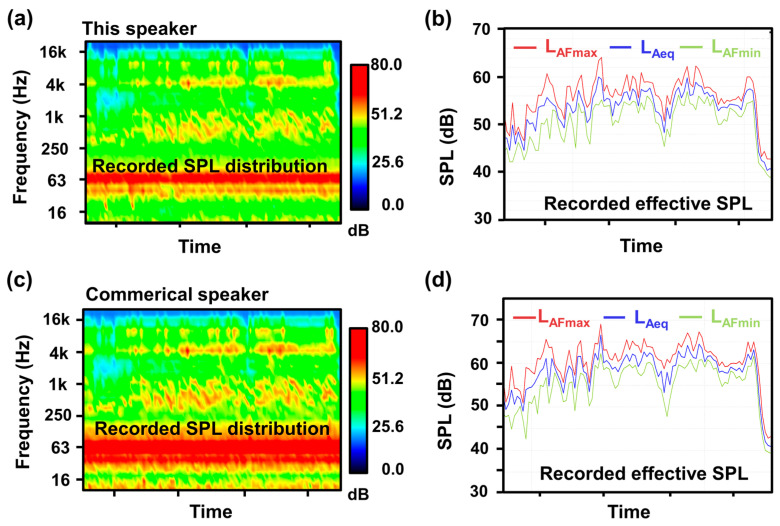
Comparison of a designed piezoelectric MEMS speaker and a commercial one. SPL distribution (**a**,**b**) effective SPL of proposed speaker at each moment when playing a song. (**c**) SPL distribution and (**d**) effective SPL of a commercial speaker at each moment when playing the same song.

**Figure 5 micromachines-14-00702-f005:**
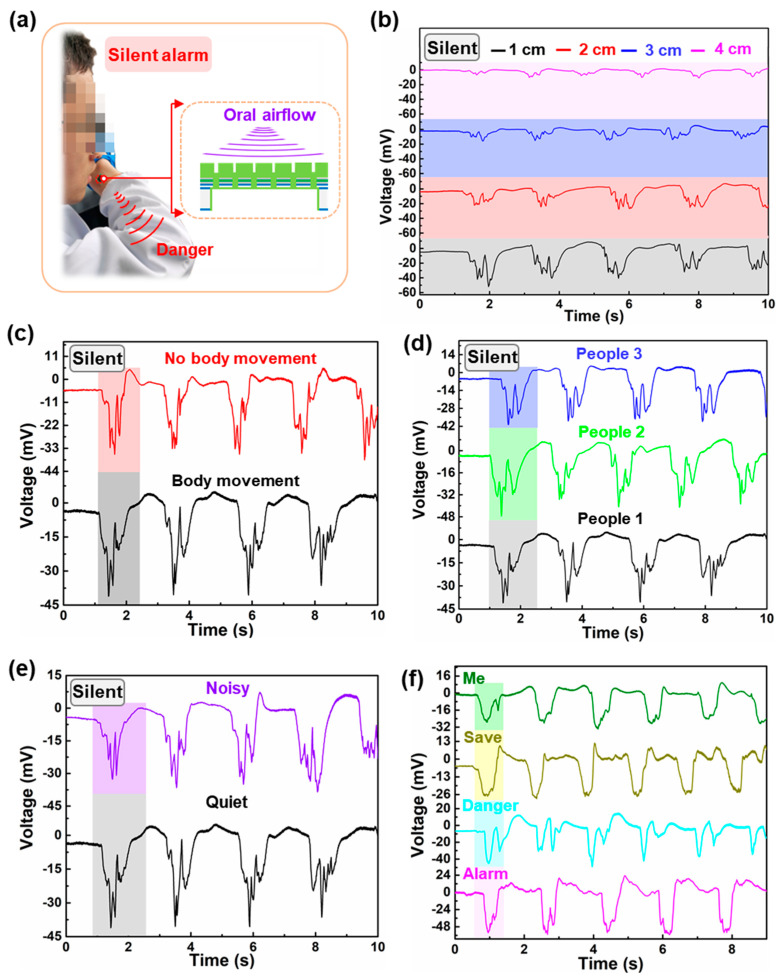
Silent alarm. (**a**) The wearing method and working mechanism. Unique voltage−signal characteristics of word “silent” (**b**) in different distances and (**c**) with and without body movement and (**d**) of different people when speaking the same word silently. (**e**) Anti−interference performance of device in a noisy sound environment. (**f**) Voltage-signal characteristics of the different words when speaking silently.

**Table 1 micromachines-14-00702-t001:** Comparison between this work and our previous work.

Paper	Structure	Radius (Side Length)	Main Supporting Layer	Thickness of Parylene C	Role of Parylene C	Minimum SPL (20 Hz-20 kHz @ 2 V)	Appilications
[10]	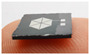 Cantilever beam array	2 mm	Si	0.5 μm	Close the gap	~59 dB	Speaker
[33]	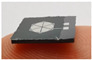 Improved cantilever beam array	2 mm	Si	0.5 μm	Close the gap	~55 dB	Speaker,motion monitoring, health warning
[36]	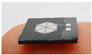 Cantilever beam array	2 mm	Si	0.2 μm	Isolate water vapor	No	Silent speech recognition
[32]	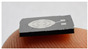 Central circle surrounded by concentric double rings	1.5 mm	Parylene C	10 μm	Supporting layer	~52 dB	Speaker
This work	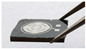 Central circle surrounded by concentric double rings	2 mm	Parylene C	10 μm	Supporting layer	~62 dB	Speaker,Silent alarm

## Data Availability

Not applicable.

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
