# Peer review of "A Piezoelectric MEMS Speaker with a Combined Function of a Silent Alarm"

_micromachines, 2023, doi:10.3390/mi14030702_

Round 1

Reviewer 1 Report

This paper presents a piezoelectric MEMS silent alarm sensor that can also function as a speaker. The silent alarm sensor can perform well in recognising words according to their unique voltage signal characteristics, and avoid some problems faced by conventional sensors. The obtained SPL of the proposed device can improve the low frequency performance. This paper can be considered for publication after addressing the following minor issues: 

1) Why do you choose the central circle surrounded by concentric double rings? and what is the advantage of the rigid-flex structure compared to others?

2) Regarding the PZT thin-film, what is the deposition method and what are the material properties (i.e. dielectric constant, Young’s modulus)? and do you need to polarise the PZT layer before testing the output, if so how did you do this, if not why?

3) Figure4 gives the measured SPL for the two structures of rigid-flex coupling and rigid supporting layer uder 20k Hz. For rigid-flexible coupling structures, the piezoelectric element undergoes a larger rigid displacement due to the deformation of the device, but its strain may be small. What is the amplitude of the piezoelectric output for the two structures tested in Figure4 under the same sound pressure excitation ranges from 20 to 20k Hz?

Reviewer 2 Report

The authors published a series of papers on the same type of device: 10.1016/j.nanoen.2021.106324, 10.1109/JMEMS.2021.3087718, 10.1109/MEMS51670.2022.9699830, 10.1109/MEMS51670.2022.9699681, it is unclear what the novelty is in this manuscript, compared with their previous publications.

Reviewer 3 Report

Paper presents the design and characterization of a MEMS piezoelectric resonator, proposing potential use in applications featuring devices like silent alarm sensor and speaker.
The topic is of interest to the broad audience, it fits into the scope of the MDPI Micromachines journal and poses a potential impact in the field. It is written in a decent English and in sufficient details...
However, there are some drawbacks in writing and presenting results, related to the scientific soundness.  

Title is a kind of misleading.
We expect reading about the application of a piezoelectric MEMS resonator, but it is actually about its fabrication and characterization and the fabrication and characterization thermselves are not as much new as authors claim ("ingenious" as they say in Introduction and Conclusion). On multifunctional devices based on MEMS resonators has been written a lot.
Parylene C as a constitutive element in piezoelectric composites is not new, too.

Authors themselves have history of results related to piezoelectric MEMS resonators and there is an understandable overlap in paper production (10.1109/JMEMS.2021.3087718 2021, 10.1016/j.nanoen.2021.106324 2021).
In order to see every contribution as unique, we need to see whatr is genuinly new.

Major remark is about focusing on what is genuinly new: material, structure or application and on elaborating the each one of chosen aspects, throughout the whole text, abstract, introduction, methods, results, discussion and conclusion.
With sufficient details.
For instance: dual function. It is the eye catcher in the title but throughout the text we do not see if the resonator is intended to operate in an array where functions are separate and how, where is a part on signal conditioning, circuit design with additional parts (bluetooth module etc).
If functions are exclusive (which might be obvious in another context) then some more elaboration on the way they are utilised in a specific application would bring added value to the text because title suggests it is the main contribution of this research.

Apart from this objection related to the very concept, and the request for the explanation of what is genuinely new and how it can be of service to the readership (with sufficient details), I have no minor remarks, I recommend a major revision of the whole text.

Thank you for your contribution and your diligent work. It was engaging to read it.

With kind regards, 

Round 2

Reviewer 2 Report

The authors demonstrated a dual-functional device that has improved performance compared to their previous work. The device has been fabricated with good quality, and it was measured and tested with various methods. The manuscript is ready for publication after the following minor issues are addressed:

1. There is no 'section 4', and the conclusion part should be labelled with the number 4, not 5.

 2. Figure 2. It seems that the authors reused some of the schematics from their previous publication. Please consider replotting with a different style, or get permission to reuse from the other publisher.

3. The subplots in Figure 5 seem to have wrong y-axis values. For example, in Fig. 5d, it's weird that people-1's voice generates a negative voltage while the other two generate positive voltages.

Reviewer 3 Report

Thank you for your additional effort.

Some typos remained:

Discussion should be paragraph 4 not 3.

Sections 3.1 and 3.2 should be re-numerated in that manner too.

With kind regards...
